# Conceiving a Digital Twin for a Flexible Manufacturing System

Laurence C. Magalhães [1,†], Luciano C. Magalhães [2,†], Jhonatan B. Ramos [1,†], Luciano R. Moura [1,†], Renato E. N. de Moraes [1,†], João B. Gonçalves [1,†], Wilian H. Hisatugu [1,†], Marcelo T. Souza [3,†], Luis N. L. de Lacalle [4,*,†] and João C. E. Ferreira [5,*,†]

1 Department of Industrial Technology, Federal University of Espírito Santo, Av. Fernando Ferrari, 514—Campus de Goiabeiras, Vitória 29075-910, Brazil
2 Instituto SENAI de Inovação em Sistemas Embarcados, Av. Luiz Boiteux Piazza, 574, Cachoeira do Bom Jesus, Florianópolis 88032-005, Brazil
3 Department of Exact and Technological Sciences, Santa Cruz State University, Jorge Amado Highway, Km 16, Ilhéus 45662-900, Brazil
4 Department of Mechanical Engineering, The Aeronautics Advanced Manufacturing Center (CFAA), University of the Basque Country, Zamudio Tecnologic Park, 48170 Bilbao, Spain
5 Department of Mechanical Engineering, Federal University of Santa Catarina, Grupo de Integração da Manufatura (GRIMA), Caixa Postal 476, Florianópolis 88040-900, Brazil
* Correspondence: norberto.lzlacalle@ehu.eus (L.N.L.d.L.); j.c.ferreira@ufsc.br (J.C.E.F.)
† These authors contributed equally to this work.

**Abstract:** Digitization and virtualization represent key factors in the era of Industry 4.0. Digital twins (DT) can certainly contribute to increasing the efficiency of various productive sectors as they can contribute to monitoring, managing, and improvement of a product or process throughout its life cycle. Although several works deal with DTs, there are gaps regarding the use of this technology when a Flexible Manufacturing System (FMS) is used. Existing work, for the most part, is concerned with simulating the progress of manufacturing without providing key production data in real-time. Still, most of the solutions presented in the literature are relatively expensive and may be difficult to implement in most companies, due to their complexity. In this work, the digital twin of an FMS is conceived. The specific module of an ERP (Enterprise Resources Planning) system is used to digitize the physical entity. Production data is entered according to tryouts performed in the FMS. Sensors installed in the main components of the FMS, CNC (computer numerical control) lathe, robotic arm, and pallet conveyor send information in real-time to the digital entity. The results show that simulations using the digital twin present very satisfactory results compared to the physical entity. In time, information such as production rate, queue management, feedstock, equipment, and pallet status can be easily accessed by operators and managers at any time during the production process, confirming the MES (manufacture execution system) efficiency. The low-cost hardware and software used in this work showed its feasibility. The DT created represents the initial step towards designing a metaverse solution for the manufacturing unit in question, which should operate in the near future as a smart and autonomous factory model.

**Keywords:** flexible manufacturing system; digital twin; simulation; Industry 4.0

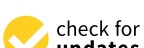



## 1. Introduction

The new generation of integrated manufacturing systems (IMS) consists of the collaborative integration of humans, cyber systems, and physical systems [1]. Humans are the creators and users of the physical system. The physical system is the main part, which is the provider of the manufacturing process. The cyber system is the core; it can analyze, calculate, and control the manufacturing process.

The human-cyber-physical system (HCPS) represents an innovation archetype brought by the new generation of artificial intelligence technology in the intelligent manufacturing

sector [2]. It demonstrates the deep integration and continuous upgrade of the cyber-physical system (CPS) and human-cyber system [3]. A basic component of Industry 4.0 is the flexible manufacturing system (FMS), an advanced production system that interconnects machines, workstations, and logistics equipment, the entire manufacturing process being coordinated with the computer [4].

In industrial applications, flexible manufacturing systems (FMS) are the most important, automated, and technologically sophisticated of the machine cell types used to implement cellular manufacturing with advanced technology. An FMS is essentially a computer-controlled production system, which brings together different tools, machines, and control equipment capable of processing a variety of part types or jobs. The FMS concept integrates many advanced technologies, including flexible automation, CNC (computer numerical control) machines, distributed computer control, and automated material handling and storage. Some advantages of FMS include: improved capital/equipment utilization, reduced work-in-progress and set up, substantially reduced throughput times/lead times, reduced inventory and smaller batches, and reduced manpower [5]. Thus, emerging technologies are bringing new approaches to manufacturing parts and products, employing distributed and collaborative production. In this context, simulation systems and software are also very much used. Through these tools, it is possible to simulate business systems and manufacturing processes by analyzing system input and output in real-time and obtaining a detailed report about the process under study [6].

The implementation and commissioning of a manufacturing system and, in particular, an FMS, involves high expenses, and the expected savings are difficult to specify. For this reason, both during the design of the structure of the flexible manufacturing system and in its operation, modeling and simulation techniques are used [4]. A virtual simulation model reflects the constraints of physical assets without errors and direct training on physical assets to extract appropriate solutions through virtual simulation [7]. As such, it is an efficient tool for extracting appropriate actions for physical assets, being more advantageous than performing the trial-and-error method, which is more expensive and time-consuming. Thus, it is possible to solve problems with physical assets in advance through 65 experiments [8,9].

Nevertheless, because virtual simulation does not involve linking data between the physical and cyber worlds, it cannot actually generate a "twin". Therefore, virtual simulation cannot be regarded as digital twin [10]. Digital twin (DT) creates the virtual model of a physical entity in a digital way, promotes the interaction and integration of the physical and digital worlds, and builds a reliable bridge for industrial information integration. DT has been evolving at a fast pace, being widely used in smart manufacturing [11]. DT can link the physical and cyber worlds to enable the optimization of the entire manufacturing process [12].

According to a recent report by Grand View Research, the size of the global DT market was valued at USD 7.48 billion in 2021 and is expected to grow at a compound annual growth rate (CAGR) of 39.1% from 2022 to 2030. The integration of DT technology with other technologies such as the Internet of Things (IoT), artificial intelligence (AI) and cloud computing is expected to drive market growth [13].

The contributions of this paper are as follows: (a) build a digital twin for a flexible manufacturing system consisting of 2 robotic manipulators, 1 machining unit, 1 CNC mill and 2 conveyor belts, based on the ISO reference model. (b) Generate a real application that allows investigating the accuracy of the information generated by the DT compared with that provided by the FMS. (c) Use the signals from the sensors available in the FMS to provide information to the DT in order to support decision making. (d) Verify the possibility of using the CIMSoft V 88-113D Amatrol software (Jeffersonville, IN, USA) and its effectiveness for constructing the DT and integrating with the cyber-physical system.

*1.1. Background of Digital Twins: Concept and Integration Levels*

The term "Twin" was initially originated at NASA (National Aeronautics and Space Administration), through the simulated environment developed during the Apollo 13 mission, which consisted of building two identical space vehicles. While one spacecraft was launched into the air to carry out its mission, the other remained on the ground, allowing engineers to mirror the launch conditions [14]. Later, with technical improvements, the digital twin (DT) was adopted in the aerospace industry by NASA and the North American Air Force [15]. Consequently, the spacecraft kept on the ground was replaced by the digital replica to provide more information through high-fidelity simulations [16]. The DT initially consisted of three dimensions, including a physical part, a digital counterpart, and a connection for communication between the two parts [16,17]. In 2010, NASA illustrated the definition and role of DT for space vehicles in detail in the Draft Modeling, Simulation, Information Technology and Processing Roadmap [18]. In the following year, the United States Air Force (US Air Force) developed the application of DT in the monitoring and management of the structural health of the aircraft [19]. In 2012, NASA and the US Air Force jointly published an article on DT, which stated that DT was the key technology for future vehicles [15]. After this publication, the amount of research on DT in the aerospace industry has grown continuously. For example, Reifsnider et al. [20] presented a multiphysics-stimulated DG simulation methodology for fleet management. According to Mandolla et al. [21], DTs are able to moderate damage or degradation, activating a self-healing mechanism or 110 recommending changes in the mission profile, with the aim of reducing loads. One of the first DT works in the manufacturing area was proposed by Lee et al. [22], as the virtual counterpart (stored in the Cloud) of production resources in the context of Industry 4.0. In 2014, the first white paper referring to the DT was published and the three-dimensional structure of the DT was widely disclosed [23]. Subsequently, DT was introduced in several areas besides the aerospace industry, such as oil and gas and medicine [16]. As of 2016, the term "Digital Twin" has been growing rapidly and becoming a "hot" topic among researchers, educators, software developers and professionals in the field [24]. In 2017 and 2018, Gartner deliberated DT as one of the ten technology trends for the next decade [25].

A DT can be defined as a digital model that reflects, in a timely manner, the state of a corresponding twin system based on historical data, real-time sensor data, and physical model [15]. A DT can mirror one or more resources of a physical environment, such as operators, machines, materials, and environments. The digital part is composed of functionalities for the purposes of data management, analysis, and computation [26,27]. A DT is more than a simulation model [14,23,28,29]. It is an intelligent and evolving model, being the virtual counterpart of a physical entity or process. It follows the life cycle of its physical twin to monitor, control and optimize its processes, continuously predicting its future states [10]. In general, DTs are useful for reducing production costs, improving and streamlining the test and production cycle of a product or process, reducing the time to introduce new products to the market, and creating a virtual environment in which all phases of the product are integrated [23]. The DT can add value to companies and many new business models can emerge from it [30]. Although there are several definitions of DT, one of the most referenced (focused on NASA's autonomous space exploration vehicles) is given by [15]: a DT is an integrated, multiphysics, multiscale and probabilistic simulation of a system, which uses physical models and data to mirror it reliably and in real time, predictively taking care of its own permanence in the environment in the face of unforeseen problems in the real or virtual system. Based on this definition, researchers from different universities and institutes proposed their own understanding of DT [16]. It is clear that the concept of DT depends on the context and on the point of view necessary for the specific application area, that is, the DT is an adequate digital representation. In this way, a DT only needs to collect data relevant to the use of interest, instead of all possible and available data from the physical system [31]. With the advancement of research on DT, and the perception of the importance of this concept for the achievement of intelligent

manufacturing, its adoption in industries is becoming increasingly beneficial. Among these benefits, we can mention greater visibility of the business, agility in product development, process optimization, and improvement in service quality [16]. However, as in the literature, different understandings of DTs can be observed in the industry.

According to Kritzinger et al. [29], many works found in the literature are not really DTs, as there is no bidirectional communication between the DT and the physical system. There are three levels of virtualization for a given system, as illustrated in Figure 1. (a) Digital Model: the simple digital representation of a physical object. There is no interaction between the physical and virtual models; (b) Digital Shadow: characterized by the connection between physical and digital objects, but in a unidirectional way, with the physical object updating the digital object; (c) Digital Twin: also characterized by the connection between physical and digital objects, but in a bidirectional way, with the physical object updating the digital object, and vice versa.

The transformation of industrial equipment into a CPS and its subsequent virtualization in the form of a DT is not a trivial process [10]. It involves the integration of different systems, the interoperation of different protocols and communication formats, the analysis of suitable protocols to support the transmission of large volumes of data in real time, the construction of wrappers on top of legacy PLCs, the synthesis instrumentation, and IoT and data from industrial networks, among other aspects [32,33]. There are other entities in the environment with which a DT must interact and for which appropriate forms of communication must be designed [10]:

- Between physical and virtual twins;
- Between the DT and other different DTs in the surrounding environment;
- Between the DT and domain experts, who interact and operate in the DT, through human interfaces.

In terms of the functionality of a DT, Tao et al. [34] and Cimino et al. [34] list the following:

- Real-time monitoring to update the DT, with information about parts, products, operations, machines, and all entities that make up the physical environment;
- Analysis and forecasting of energy consumption;
- Smart optimization and updating, based on user operation analysis and product and/or production process data;
- Analysis and user operation behavior, to detect and evaluate the operations performed by him;
- Virtual maintenance of the product, using virtual or augmented reality, for testing, evaluation and, finally, for real maintenance;
- Failure analysis and forecasting for equipment and/or product maintenance planning.

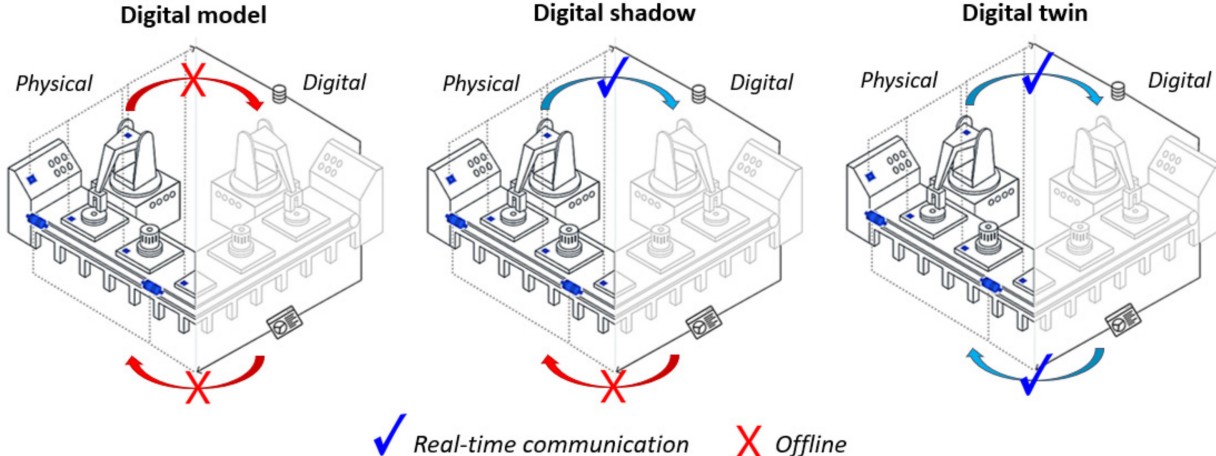

**Figure 1.** Levels of virtualization.

The functionalities of a DT consider the application domain and the aspects to be evaluated and visualized. Thus, there may be a variation in the level of detail that a graphical interface must have, as well as the level of performance in the real system [29]. Stark et al. [35] proposed a general model of a DT, dividing its functionalities into two broad categories: Context & Environment, which cover functionalities linked to the physical object or process and its connectivity; and Behavior & Capability, which involve the intelligence of the digital model, the 3D fidelity of the physical object, simulations, and interactions with users.

### 1.2. A Brief Literary Overview of Digital Twins for Manufacturing Systems

Many authors have proposed digital twins for manufacturing systems [36–39]. Botkina et al. [36] developed a DT of a real cutting tool, which contains the data format and structure, and information flows. Then, the DT is stored, refined, and propagated to the process of planning for an optimized machining solution. Cai et al. [37] present an integration of manufacturing data and sensory data in the development of "digital twins" virtual machine tools to improve their accountability and capabilities for cyber-physical manufacturing. The sensory data were used to extract the machining characteristics profiles of a DT machine tool, with which the tool can better reflect the actual status of its physical counterpart in its various applications. An example of developing the DT of a 3-axis vertical milling machine was presented to demonstrate the concept of modeling and building a digital twin's virtual machine tool for cyber-physical manufacturing. The presented technique can be used as a building block for cyber-physical manufacturing development.

Lohtander et al. [38] conceived the DT of a micro manufacturing unit (MMU). Overall, the DT consists of a machine, material, method, measurement, and modeling fields, like a physical world describes. The dynamic model of a machine tool DT was developed by Scaglioni and Ferretti [39]. The main features of the model are the FEM-based description of the structural flexibility of the components of the kinematic structure, the model of the cutting process, and the model of the transmission chains and control systems. The model was validated with respect to the motion dynamic behavior.

Duan et al. [40] created a blade-rotor test rig DT system to solve the problems of low visibility and poor equipment-monitoring capabilities. The process was mapped to the virtual space in real-time, which had a strong ability to control the operating state of the equipment. In this case the DT not only assured the safety of personnel under the high-speed operation of the equipment, but also realized the dynamic test process of the equipment. With visual monitoring, through the data interactive panel designed in the virtual space to achieve the monitoring of the data, technicians can make timely feedback to the physical world by sensing data abnormalities, ensuring the safety of the test equipment.

Zhao et al. [41] proposed a surface roughness stabilization method, based on DT-driven machining parameters self-adaption adjustment, to obtain stable surface quality in five-axis machining. A novel self-learning surface roughness prediction model based on Pigeon-Inspired Optimization and Support Vector Machine (PIO–SVM) was proposed, which takes the influence of cutter posture and cutting force on surface roughness into consideration. Lead and tilt angle and spindle speed were selected as the adjustable parameters. Using the proposed method, the surface roughness of the workpiece became stable and met the final machining requirements, which greatly improved the machined surface properties, machining efficiency, and reduced manufacturing cost. This is of crucial importance for the machining of complicated and precise workpieces with sculptured surfaces.

Tong et al. [42] presented a real-time machining data application and service based on an Intelligent Machine Tool (IMT) digital twin. Multisensor fusion technology was adopted for real-time data acquisition and processing. Multiple forms of human-machine interfaces and applications were developed for data visualization and analysis in DT, including the machining trajectory, machining status, and energy consumption. The DT model was established with the aim of further data analysis and optimization, such as machine tool dynamics, contour error estimation, and compensation. Results showed that the IMT digital

twin system provides a comprehensive platform for manufacturing process monitoring and optimization.

Qiao et al. [43] presented a data-driven model for DT, along with a model prediction method based on deep learning that uses a technique to predict the condition of a machining tool. Studies using vibration data measured on milling tools showed the effectiveness of their DT model for predicting tool wear. Ward et al. [44] described a machining DT on a large-scale CNC machine tool with adaptive control of machining operations integrating real-time model-based simulations with online feedback for residual stress control, chatter prediction, and adaptive feed rate. The feedback was able to update the spindle speed and feed rate in real-time to prevent chatter before it occurred. This method showed that vibration could be predicted and avoided, thus protecting the tool, the workpiece, and the machine.

Xie et al. [45] developed a DT to track the life cycle of a cutting tool. They developed a virtual cutting tool testing platform with physical and virtual tool wear data to ensure the continuous improvement of the process and tool. Zhuang et al. [46] proposed a method based on DT to monitor in real-time the wear of 264 cutting tools for turning operations. According to the authors, their DT model did not consider tool nose eccentricity and room temperature variation, nor the tool life cycle, which includes product design, manufacturing, and service phases. More DT models need to be developed which seek to improve the accuracy of tool life prediction algorithms.

López-Estrada et al. [47] presented a DT of a single-edge micro-cutting ma chine tool in a collaborative cloud-based product lifecycle management (PLM) platform. The DT was designed to simulate machine behavior under different cutting process conditions. The authors also sought to integrate models created by different computing applications into a single collaborative PLM platform and obtain an interoperable model-based DT. According to the authors, the integration of existing models is complex and requires a significant amount of rework. Additionally, the integration of the kinematic and dynamic models required redefining many of the constraints and data included in both models.

Fan et al. [3] described a general DT visualization architecture for flexible manufacturing systems (FMS) in order to address the life cycle functionality of human-cyber-physical manufacturing systems. The authors showed how CP DT modeling of heterogeneous information can be described and how to explore human-machine interaction visualized in 3D with DT scenario information. For this, the authors propose a modeling concept called GHOST (geometric information, historical samples, object attribute, snapshot collection and topology constraint) for the development of prototypes.

Redelinghuys et al. [48] presented a multi-layer architecture that provides the infrastructure required for a DT within the cyber-physical production systems (CPPS) paradigm. Even though the paper specifically considers a manufacturing case study, the architecture is independent of the application-specific details and facilitates wider application. The architecture provides a local data layer, an IoT (Internet of Things) gateway layer that relays information between the physical world and cyberspace, a layer with cloud-based data repositories, and, finally, a layer with emulation and simulation software. In that study, a robotic gripper assembly DT was developed.

The building process of a high-fidelity DT model is divided into two parts. The mechanical equipment's transmission part is decomposed according to the meta-action theory, while the static part is decomposed according to its sub-function. The machine is decomposed into transmission chains according to FMA (function—motion—action) decomposition, and the chain is decomposed into transmission units. Each transmission unit has inputs and outputs, and has a corresponding data interface. Once the entity model is established, the sensor can then be installed at its corresponding position on the transmission unit in order to maintain data transmission between the PE (physical entity model) and VE (virtual equipment model), so that VE can carry out the mapping to PE to the greatest extent possible [49]. In this context, this article describes the concept and

simplifies the methodology in the application of a DT in a flexible manufacturing system. Table 1 presents a comparative study of the characteristics treated in digital twin literature.

**Table 1.** A comparative study of the characteristics treated in DT literature.

| Author (Year) | Monitoring | Sensors | Simulation | Scheduling | Real Time | Real FMS |
|---|:---:|:---:|:---:|:---:|:---:|:---:|
| Cai et al. (2017) [37] | ✓ | ✓ | | | | |
| Botkina et al. (2018) [36] | | | ✓ | | ✓ | |
| Lohtander et al. (2018) [38] | | | ✓ | | ✓ | |
| Scaglioni and Ferretti (2018) [39] | | | ✓ | | | |
| Qiao et al. (2019) [43] | | ✓ | | | | |
| López-Estrada et al. (2019) [47] | ✓ | ✓ | ✓ | | | |
| Zhao et al. (2020) [41] | | ✓ | ✓ | | ✓ | |
| Tong et al. (2020) [42] | ✓ | ✓ | ✓ | | ✓ | |
| Redelinghuys et al. (2020) [48] | ✓ | | ✓ | | | |
| Ward et al. (2021) [44] | ✓ | ✓ | ✓ | | ✓ | |
| Duan et al. (2021) [40] | | ✓ | ✓ | | | |
| Xie et al. (2021) [45] | ✓ | ✓ | ✓ | | ✓ | |
| Zhuang et al. (2021) [46] | ✓ | ✓ | ✓ | | ✓ | |
| Fan et al. (2021) [3] | ✓ | ✓ | ✓ | | | |
| Yang et al. (2022) [49] | ✓ | ✓ | ✓ | | ✓ | ✓ |
| This paper | ✓ | ✓ | ✓ | ✓ | ✓ | ✓ |

Considering the aforementioned scenario, this work also seeks to improve tool control decisions within a specified work cell. The practical implementation of the proposed concept is carried out in an FMS, comprising one robotic arm, a CNC lathe machine and a conveyor. The proposed DT has been tested using the CIMSoft V 88-113D Amatrol tool (Jeffersonville, IN, USA), which is an effective and accurate computer-based software tool to support the digital transition towards the implementation of Industry 4.0. Finally, a comparison is made between real robotic and simulated operations. Before presenting the methodology for designing the DT described in this work, the potential and perspectives of combining this technology with the metaverse are pointed out.

### 1.3. Integration of Digital Twins with the Metaverse

Digital twins will be one of the essential building blocks of the metaverse [50,51], composing virtual representations of objects or systems in the digital world [52]. The integration of DTs and the metaverse can help recreate digitally real-world existence. As long as the metaverse components are fed with real and accurate data, what-if scenarios can be run successfully. In these scenarios, simulations and machine learning will be used to help with decision making. DTs and simulation technology will allow the metaverse to support remote maintenance of machines that need to be serviced and potentially connected or mapped to a real machine. The metaverse will allow humans to dynamically interact in environments composed of digital images, but without the ability to change the state of those objects. The convergence of these technologies will allow humans to operate machines, make mistakes without consequences when experimenting with new configurations, and learn from any aspect of a DT environment [53–55].

A physical space can be captured and transformed into an accurate and immersive digital replica using 3D spatial data technology and simulation technology, giving rise to a "metafactory". Digital worlds based on real-time data will allow corporations to have multiple iterations and simulations of themselves. This feedback from the digital world will allow companies to change real-world parameters to optimize processes. In manufacturing industries, both technologies can be used to create virtual copies of entire factories to easily identify faults and visualize the production process [56]. With this capability, manufacturing industries can design, manufacture, process, and maintain the production process more efficiently. In industrial IoT establishments, DTs in the metaverse can be used to help monitor, track, and control systems before they are permanently deployed. As the machine provides real-time data, the outputs can be used to check possible system failures [53,54].

Another issue is the possibility of circumventing the lack of qualified engineers. The metafactory will allow distant workers to immerse themselves in the twin and interact with settings and data in real time. Metafactories will also immerse people in sophisticated

training. The experience would be as realistic as the flight simulators used in pilot training. In addition to training people to be less prone to making mistakes, the metafactory would provide accurate feedback to resolve the design and training issues that caused those mistakes [57].

Thus, the integration of DTs and metaverse technology has high potential to improve the processes of many industries, enabling better forecasting, monitoring, tracking, allocation, resource management, optimization, and quality control. Furthermore, the implementation of technologies such as virtual reality (VR), augmented reality (AR), machine learning, and blockchain may lead to accurate predictions [54].

## 2. Proposed Methodology

The implementation of the proposed methodology is described in this section.

### 2.1. Reference Architecture of the Digital Twin

The reference architecture of the proposed digital twin is similar to the one described in [58], which considers the ISA-95 reference model for industrial automation [59]. The cyber-physical system (CPS) is usually linked with the network protocols of levels 0 (industrial equipment), 1 (industrial networks and instrumentation), and 2 (physical controllers such as CNC and PLC) [60], providing it with communication capabilities to exchange information with client applications such as the DT and the enterprise resource planning (ERP) module. IoT technology can replace PLCs and SCADA systems, depending on the particular factory floor. Users can interact at different levels with the CPS and its DT, as well as with DTs representing other CPS or production entities.

Levels 3 and 4 act as client systems of the CPS and the DT. Although the DT acts in a symbiotic way and as a virtual image of the physical CPS, the DT can also be seen as a client of the CPS. Similarly to [58], these systems operate in a PDCA cycle. In general, level 4 (e.g., ERP, CRM) and level 3 (e.g., MES) systems plan actions ("Plan"), levels 0, 1, and 2 (the CPS) execute the actions ("Do"), level 3 checks ("Check") whether they are being executed as planned, and levels 1, 2, 3, and 4 adjust ("Act") the actual CPS system in case of deviations from the plan.

Figure 2 shows schematically the reference architecture used for the proposed DT, whose design method is detailed in the following section.

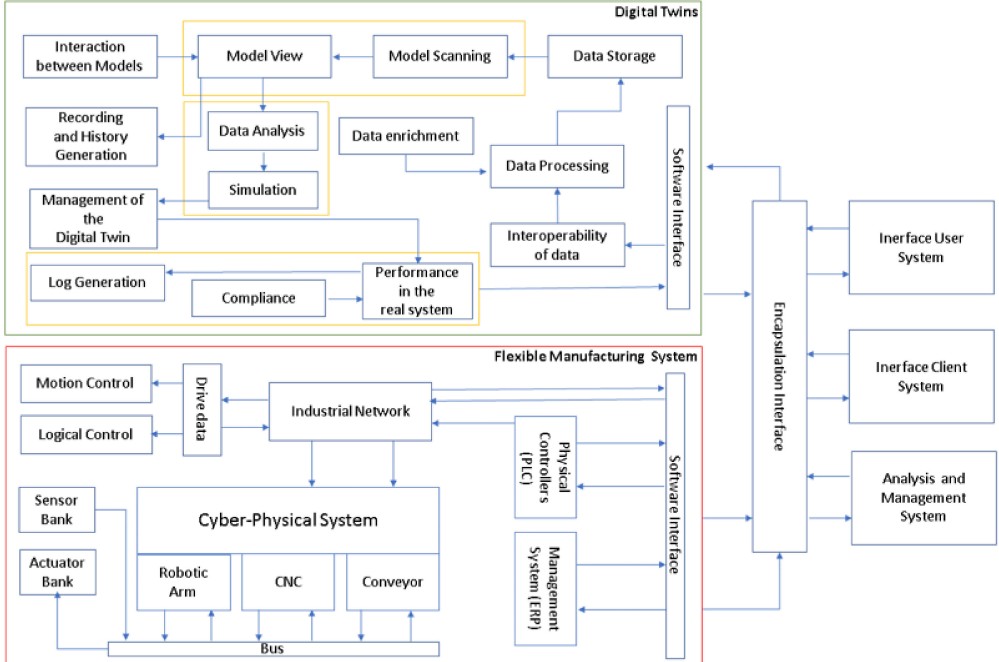

**Figure 2.** Reference architecture of the proposed digital twin.

### 2.2. Steps for Creating the Proposed Digital Twin

The DT proposed in this work was created using four specific modules of the Amatrol V 88-113D CIMSoft (Jeffersonville, IN, USA) ERP and manufacturing execution system, namely production planning, manufacturing management, facilities layout, and system simulation. The production planning module is used for creating work centers and costs in the ERP system. The manufacturing management module is used to schedule and release manufacturing orders. The facilities layout module enables digitizing the FMS components in a virtual environment that interacts with the system simulation module in real time. In the facilities layout module, the human-machine interface (HMI) is defined. HMI development software packages vary in their development methods. Typically, objects are selected from a library of pre-written graphical objects. Each object has a set of parameters that must be defined. Such parameters may include the object's location within a coordinate system, among others. The graphical objects of the presentation are connected to a database through identifiers (tags). An identifier is an object that is used to reference a piece of data. When data within the database are updated (via the PLC or other controller), the object linked with the HMI display is updated. This update is based on the information sent by sensors, such as (a) rotary spindle speed, axis travel, tool indexer in the CNC lathe; (b) end effector torque sensor, travel speed in robot arm; (c) conveyor speed and RFID bar code sensor in pallet and conveyor. These pieces of equipment are present in FMS as shown in Figure 3. Transmission control protocol (TCP) was used in data acquisition and communication. This guarantees reliable point-to-point data transmission.

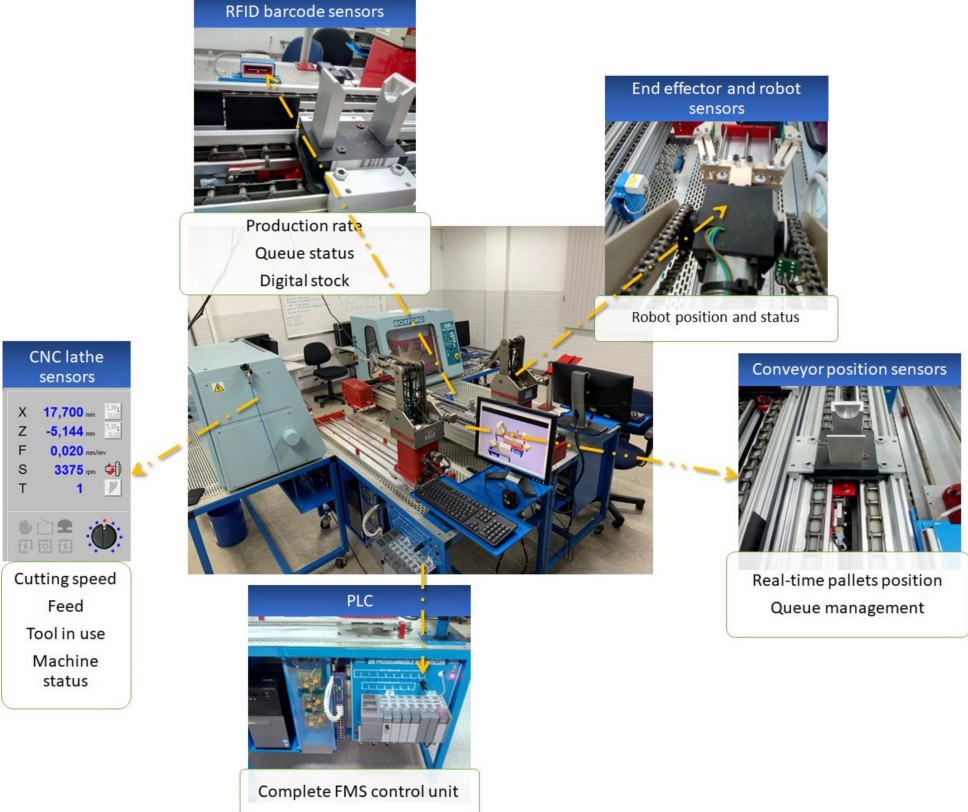

**Figure 3.** Main sensors in the FMS.

The Amatrol CIMSoft graphic library contains several graphic objects that are already connected to the relevant data. This eliminates the need to enter identifiers during HMI development. Some of the general steps for developing an HMI using the CIMSoft V 88-113D (Jeffersonville, IN, USA) software package are as follows:

- Step 1: Choose the type and size of conveyor system to use. This identifies the shape, length, and any conveyor protractor. This information is important because the production rate simulation by DT takes into account these pieces of information as well as robot displacement speed.
- Step 2: Choose the type of material storage and delivery.
- Step 3: Locate the conveyor pallet positioning stations. These correspond to the locations where the pallets will stop by workstations.
- Step 4: Identify pallet positioning stations with station number, work center ID, and programming address. This informs the system which work will be carried out and what address to use to send and receive data.
- Step 5: Add equipment to work centers. This includes CNC machines, robot arms, conveyors, part feeders, computers, etc.
- Step 6: Add descriptive texts to label stations or equipment.

These steps are summarized in Figure 4.

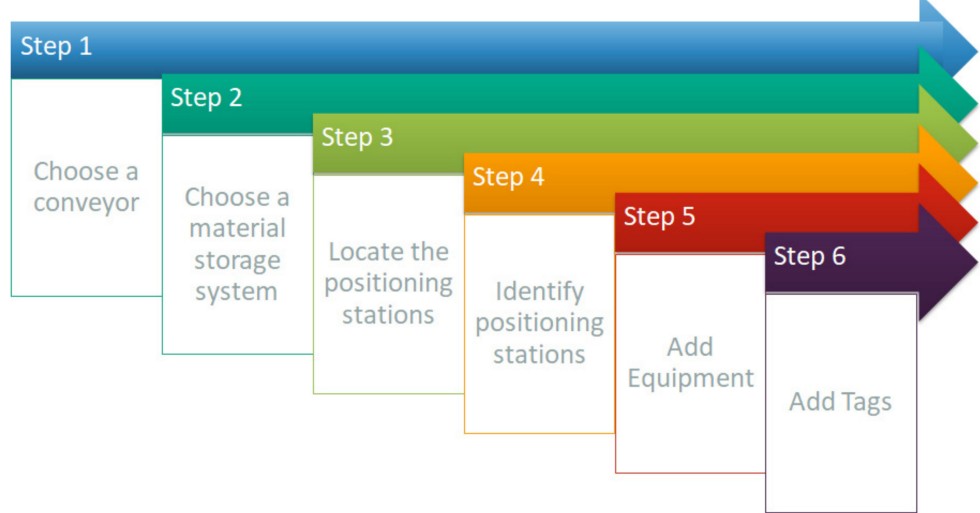

**Figure 4.** Steps to create the digital twin.

After these steps the user can then run a manufacturing simulation. At this point, the following steps should be observed: (a) verify that all data have been entered: bill of materials (BOM), identified work centers, completed and released manufacturing orders, created and saved HMI layout; (b) open an HMI layout file: the layout must match the route that is to be simulated; (c) add stock: supply virtual raw material; (d) select the manufacturing order (must correspond to HMI); (e) start the simulation. In this work, time and cost information were obtained from production tryouts in the real FMS. A batch of five parts was processed and average times were assigned to each work center. Table 2 shows the average values that were used in each step. These are important data to check the DT's precision during production rate simulation.

**Table 2.** Operation times of the real FMS to compare with DT simulated times (average of five turned parts).

| Activity | Description | Time (s) |
|---|---|---|
| 1 | CNC open door | 5 |
| 2 | Pick up billet on the pallet | 18 |
| 3 | Open chuck | 5 |
| 4 | Place billet, close chuck | 17 |
| 5 | CNC close door | 5 |
| 6 | Send command to the machine-cycle start | 4 |
| 7 | Machining time | 32 |
| 8 | Machine referencing for next cycle | 4 |
| 9 | CNC open door | 5 |
| 10 | Place part on the pallet | 22 |
| | Total (min) | 1.95 |

Five parts with 20 mm in diameter and 55 mm length were produced in aluminum 6061 billets. One cutting pass with a 0.2 mm depth of cut was performed on each part. Table 3 presents cutting parameters used to produce the parts. Cermet inserts ISO code DCGT090202N-SC, uncoated, by Sumitomo® were used. Machining was carried out dry. A Taylor Hobson® roughness meter model Surtronic 25 (Leicester, UK), was used to measure the roughness of machined parts, using a cut-off value of 0.8 mm and a total evaluation length of 4 mm, in accordance with ISO 4288. This is important information about process quality capability.

**Table 3.** Cutting parameters.

| Cutting Speed (m/min) | Feed (mm/rev) | Depth of Cut (mm) | Condition |
|---|---|---|---|
| 250 | 0.025 | 0.2 | Dry |

The FMS is composed by the following equipment: (a) a CNC Boxford® lathe model 160 TLC (Halifax, UK); (b) a machining center, Boxford® model 190 VMC (Halifax, UK); (c) two robot arms Amatrol® Pegasus (Jeffersonville, IN, USA); (d) two conveyors; (e) Allen Bradley® PLC (Milwaukee, USA).

The physical system is composed of (a) four support tables; (b) five computers and four monitors; (c) one PLC; (d) two CNC machines; (e) two robotic arms with spindle for transfer; (f) one raw material feeder; and (g) one conveyor accompanied by two pallets with sensors. The physical layout of the FMS was measured from a starting point (0,0,0) in three-dimensional coordinates. All these parts were inserted at layout facilities module. Figure 5 shows some definitions in the ERP system.

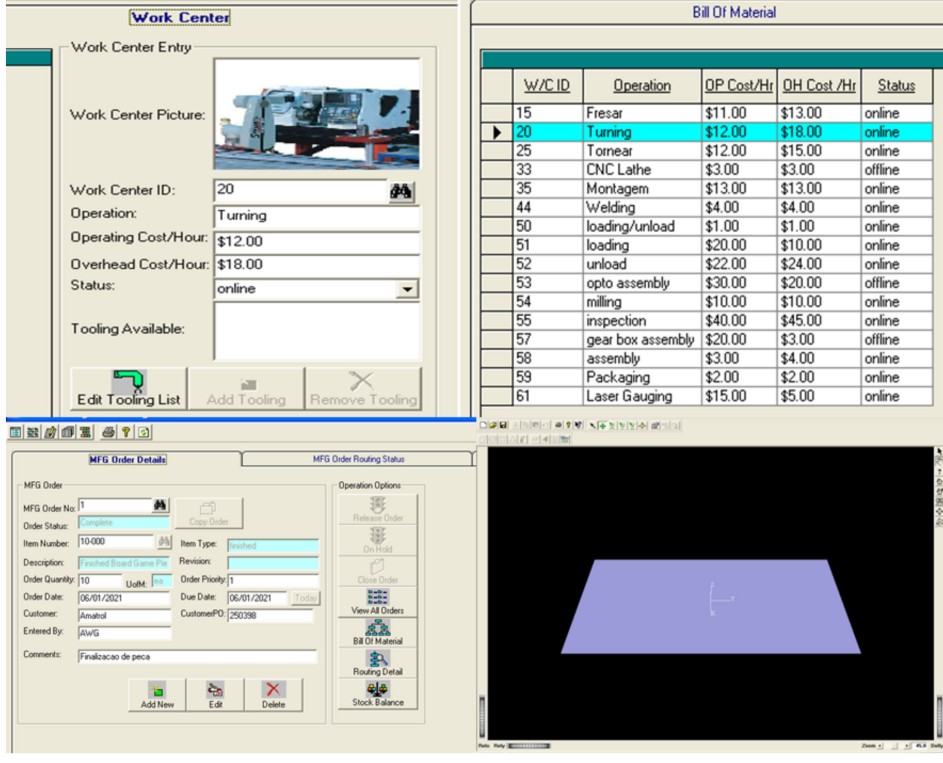

**Figure 5.** Definitions in ERP system: top to bottom, left to right: work centers creation, costs allocation, order releasing, and facilities layout ready to receive components.

Then, in order to carry out the simulation and operation of the DT, it was necessary to carry out a manufacturing order in the specific ERP system module (manufacturing management) with all the specifications required by the system: material used, material cost, processing times, stocks, and order type, and release the order on a pre-determined date.

## 3. Results and Discussion

After carrying out the steps described in the previous section, which include creating work centers and workstations, inserting the equipment and digital components into the layout facilities module, and creating and releasing manufacturing orders, the DT is fully developed and able to simulate, in real time, all the manufacturing steps performed by the FMS, as shown in Figure 6.

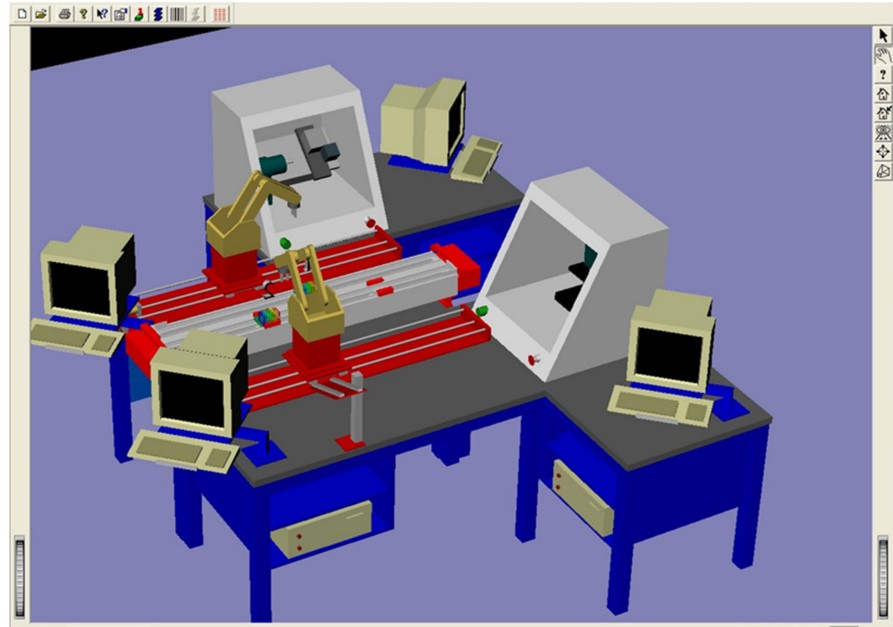

**Figure 6.** Created DT running real time production simulation.

After the manufacturing order is released and the FMS goes into operation, the DT shows, in a synchronized way, the movement of the pallets on the conveyor, the movement of the robotic arms and respective end-effectors. The status of the CNC machines is updated when they are loaded and in material removal operation. Once the first part is completely produced, the DT begins to update system information such as production rate (units/hour), production cost, queue management, cycle time per workstation, as well as the status of each station (idle or busy). This information is passed on through specific dialog boxes contained in the software and can be accessed locally or remotely, as can be seen in Figure 7.

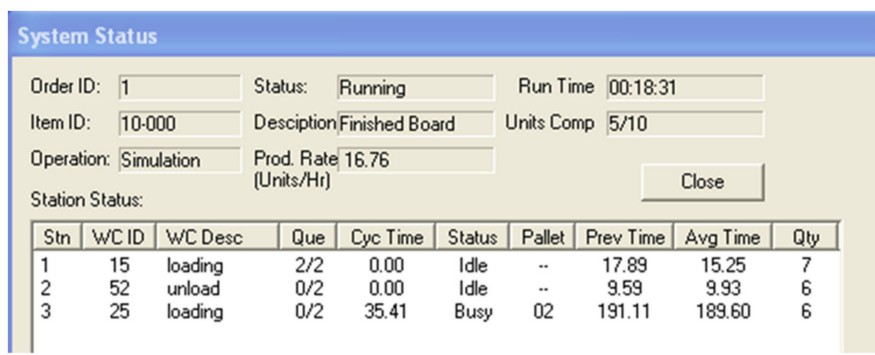

**Figure 7.** System status report.

With this information, the production control and planning staff have an overview, in real time, of the behavior of the FMS in terms of production levels. Additionally, the maintenance staff can obtain information regarding the operating status of a certain component or equipment. Once the production and simulation of a pilot batch of five parts

was completed, cycle time information from each workstation was captured and compared to the cycle times of actual FMS operation, as described in the previous section. These data are shown in Figure 8.

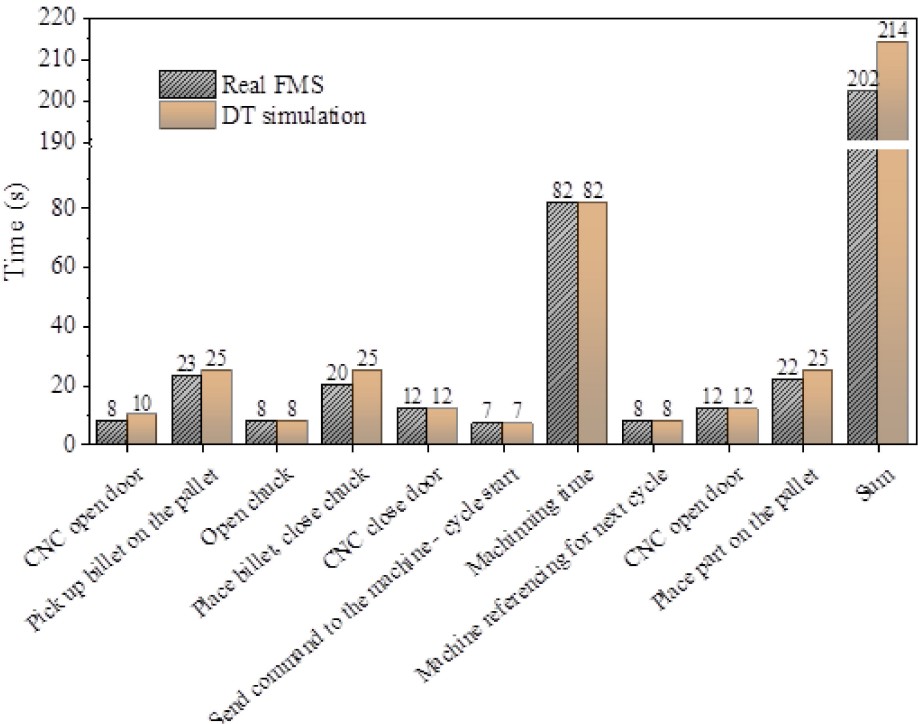

**Figure 8.** FMS manufacturing and DT simulation time.

The results show that the times simulated by the DT are only about 5% higher than the real operating cycle times of the FMS. This is mainly due to the fact that special attention was paid to filling in all the information in the facilities layout module, according to what was actually found. For example, the lengths of the conveyors had a tolerance of less than 2 mm. Once the average of five parts was used, information such as the average speed of the robots, for example, was very close to what was expected by the simulation with the DT. There was also a fast response regarding the information update by the system, showing that the data collected by the sensors are quickly and precisely passed on by the network to update the digital entity. For example, Figure 9 shows the exact moment of queuing, by the pallets, at the lathe station.

The arithmetic mean roughness ($R_a$) obtained in the processed parts was 0.25 μm, highlighting the excellent surface finish that can be achieved with the proposed level of equipment and automation, in accordance with the tolerance degree of ISO IT2-IT3. Figure 10 shows a manufactured (turned) part at the FMS.

The contribution of the created DT consists in providing, in a quick and precise way, important relevant information about the physical entity at the levels of production management as well as maintenance. The architecture used, based on ISO, is efficient and, since a good part of the solution is embedded in the layout, the costs of developing the solution were reduced.

With regard to previous research works, the results of this work move one step ahead, since it provides a solution for an FMS (i.e., a manufacturing system with two or more pieces of equipment), as discussed by Botkina et al. [36] and Scaglione and Ferreti [39]. Still, the proposed reference architecture for DTs described here is easier to use due to its lower complexity, compared with proposals presented by Fan et al. [3], for example, and its application may be carried out more easily by companies.

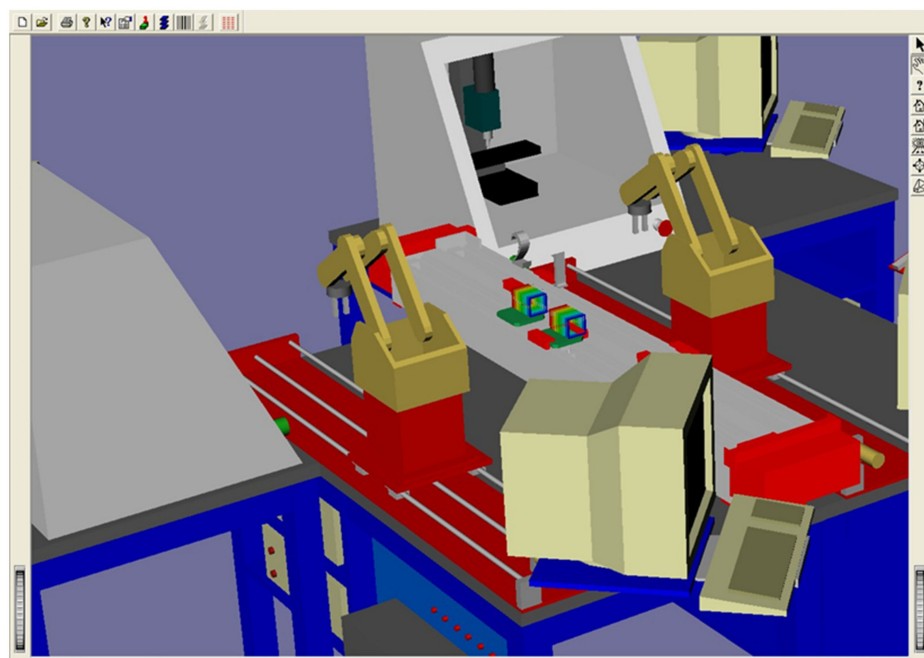

**Figure 9.** Queue formed at the lathe station.

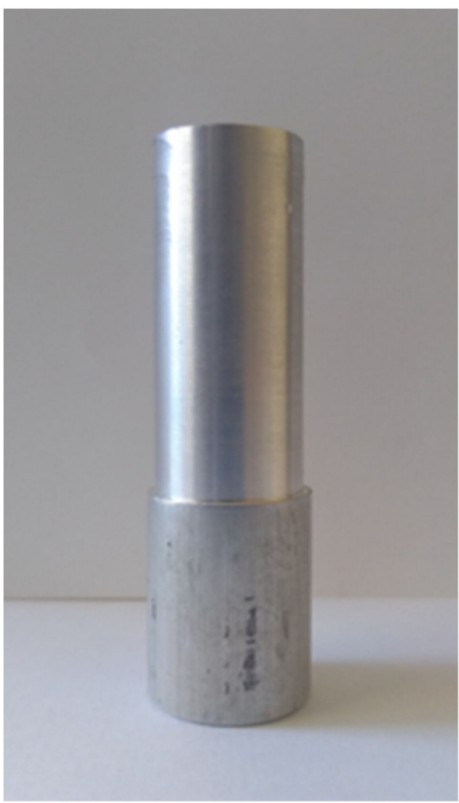

**Figure 10.** Turned part manufactured at the FMS.

It should be noticed that the objectives achieved in this work are just the initial step towards building a metaverse representing the FMS. In addition, the proposed DT solution combined with the application of technologies such as the IoT (Internet of Things), artificial intelligence (AI) and cloud computing is expected to contribute, in the future, to smart and autonomous factories.

## 4. Conclusions

In the present work, specific modules of an ERP/MES system combined with multisensors and PLC programming were used to create a digital entity capable of simulating and exchanging information, in real time, for the operation of a flexible manufacturing system composed of CNC machines, robotic arms, and pallet conveyors. The main conclusions are listed below:

- Information such as production rate, queue management, feed stock, and equipment and pallet status can be easily accessed by operators and managers at any time during the production process.
- Information can be accessed in real time at different levels of the factory, facilitating decision-making by staff. For example, pallet conveyor speed can be adjusted to improve queue management.
- The combination of the physical entity of the FMS and the digital twin created in this work with artificial intelligence, cloud computing, IOT-based communication, and augmented reality, will enable the conception of a metaverse capable of operating in a smart and autonomous factory.
- With the DT, simulations of manufacturing orders were carried out, and the results obtained were close to the actual manufacturing times in the FMS, for a small batch of parts.
- The arithmetic mean roughness ($R_a$) of the machined surfaces was equal to 0.25 μm, which corresponds to a degree of tolerance ISO IT2-IT3, i.e., with the equipment available in the FMS and the proper selection of cutting parameters, parts can be manufactured with a high degree of precision and surface finish. This shows the high capability of the process in terms of quality.
- The integrated manufacturing system composed of the FMS and the DT is an appropriate environment for training and dissemination of enabling technologies of Industry 4.0, as well as for further research in this area of knowledge. The following future works are suggested:
- Develop an application for mobile phones to monitor the simulation of the DT in real time.
- Use cloud computing to store information about the manufacturing system and make it available to operators and managers, so that they are able to make the best decisions regarding the process.
- Use an open-source process control platform to replace Amatrol CIMSoft. Such a platform should allow the integration of heterogeneous data from different devices and communication protocols.
- Develop algorithms for determining appropriate pallet conveyor speed based on queue formation.

**Author Contributions:** Conceptualization, supervision, methodology, validation, L.C.M. (Laurence C. Magalhães); simulations, programmable logic controller and robot arm programming, L.C.M. (Laurence C. Magalhães), J.B.R., J.B.G. and R.E.N.d.M.; formal analysis, L.R.M. and M.T.S.; writing—original draft preparation, J.B.R., J.B.G. and R.E.N.d.M.; writing—review and editing, J.C.E.F., L.N.L.d.L., W.H.H., M.T.S. and L.C.M. (Luciano C. Magalhães). All authors have read and agreed to the published version of the manuscript.

**Funding:** Thanks are due to Elkartek 2022 project LANVERSO, and in some sections (simulations) to Basque government university group IT 1573-22.

**Institutional Review Board Statement:** Not applicable.

**Informed Consent Statement:** Not applicable.

**Data Availability Statement:** Not applicable.

**Acknowledgments:** The authors would like to thank the Espírito Santo Technology Foundation (FEST), as well as the collaboration between the Brazilian groups and the Basque University group IT 573-22.

**Conflicts of Interest:** The authors declare no conflict of interest.

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
