# Peer review of "Conceiving a Digital Twin for a Flexible Manufacturing System"

_applsci, doi:10.3390/app12199864_

Round 1

Reviewer 1 Report

This paper presents a practical implementation of a digital twin carried out in a Flexible Manufacturing System, comprising one robotic arm, a CNC lathe machine and a conveyor.

The topic is interesting, but the paper is not suitable for publication because it lacks originality, significance and comprehensiveness in the presentation. My detailed comments are reported below:

             1) First and foremost, the topic covered by the article is very important, but it is not new, as there are many very solid scientific articles and review articles that have dealt with the same topic in the literature.

            2) The data, results and explanation presented in the article are completely insufficient to complete the content of the paper as a scientific article. The explanation of the results is completely inadequate. Also, the content of the paper is predominantly in the form of a technical report and not a research article. Perhaps this article is suitable for participation in scientific demonstrations, but in its current form, it is not suitable for publishing as a scientific article in a premier scientific journal, as it needs a precise qualitative improvement.

3) Better stress the connection between methods, linked to fundamentals, and the applied end side.

            4) The article's originality needs to be stated clearly. It is important to have sufficient results to justify the novelty of a high-quality journal paper. The Introduction should make a compelling case for why the study is useful along with a clear statement of its novelty or originality by providing relevant information and providing answers to basic questions such as: What is already known in the open literature? What is missing (i.e., research gaps)? What needs to be done, why and how? Clear statements about the novelty of the work should also appear briefly in the abstract and Conclusions sections.

5) An updated and complete literature review should be conducted and should appear as part of the Introduction, taking into account the scope and readership of the journal. The results and findings should be compared to and discussed in the context of earlier work in the literature.

Author Response

This paper presents a practical implementation of a digital twin carried out in a Flexible Manufacturing System, comprising one robotic arm, a CNC lathe machine and a conveyor. 

The topic is interesting, but the paper is not suitable for publication because it lacks originality (Deixar clara a nossa contribuição), significance and comprehensiveness in the presentation (vender melhor o artigo). My detailed comments are reported below:

             1) First and foremost, the topic covered by the article is very important, but it is not new, as there are many very solid scientific articles and review articles that have dealt with the same topic in the literature.

            2) The data, results and explanation presented in the article are completely insufficient to complete the content of the paper as a scientific article (adicionado texto da arquitetura de referência ISO para gêmeos digitais, discussão do metaverso, melhorada e ampliada a revisão da literatura). The explanation of the results is completely inadequate (Foi criada nova seção de resultados com maior foco ao que foi alcançado). Also, the content of the paper is predominantly in the form of a technical report and not a research article (Toda a seção de metodologia foi alterada deixando a parte técnica). Perhaps this article is suitable for participation in scientific demonstrations, but in its current form, it is not suitable for publishing as a scientific article in a premier scientific journal, as it needs a precise qualitative improvement. (JOAO BOSCO & LAURENCE)

3) Better stress the connection between methods, linked to fundamentals, and the applied end side. (JOAO BOSCO & LAURENCE). (A metodologia foi toda recriada para atender esse item.)

            4) The article's originality needs to be stated clearly. It is important to have sufficient results to justify the novelty of a high-quality journal paper. The Introduction should make a compelling case for why the study is useful along with a clear statement of its novelty or originality by providing relevant information and providing answers to basic questions such as: What is already known in the open literature? What is missing (i.e., research gaps)? What needs to be done, why and how? Clear statements about the novelty of the work should also appear briefly in the abstract and Conclusions sections. (RENATO)

A introdução foi redivida em seções específicas. A literatura foi ampliada. Metaverso discutido. Arquitetura de referência apresentada. A contribuição foi deixada clara ao longo do texto. Foi incluída tabela comparativa com outros trabalhos. Nosso trabalho discutido e realacionado a outros nos resultados. Abstract e conclusões reajustados.

5) An updated and complete literature review should be conducted and should appear as part of the Introduction, taking into account the scope and readership of the journal. The results and findings should be compared to and discussed in the context of earlier work in the literature. (RENATO & MARCELO)

Abstract reajustado. A introdução foi redivida em seções específicas. A literatura foi ampliada. Metaverso discutido.

Reviewer 2 Report

Dear Authors,

First of all, congratulations on the article and the research presented.

The article presents an appropriate structure and methodology for research.

The use of simulations, regardless of the field of investigation, is in my opinion fundamental for the training of good professionals.

To improve the quality of the work presented, I would suggest standardising the citation style, for example: "et al." or "et al."

In the conclusions, the limitations of the study should be included.

Best regards,

Author Response

In the conclusions, the limitations of the study should be included 

Reviewer 3 Report

This paper presents the implementation of a flexible manufacturing system and a digital twin of the system. The key components of the system include 2 CNC machines, 2 robotic arms, 2 conveyors, 1 PLC, sensors, and a number of computer. Amatrol software systems are used to programme the physical components and to create the digital twin (CIMSoft). It seems the major contribution of the paper is to prove that Amatrol CIMSoft is a good tool for creating digital twins. This can hardly be considered as a contribution.

The major weakness of this paper is that the novelty is not clearly stated, e.g. the problem (or challenges) that the paper is trying to address, the major works that have been published, the major difference between these existing works and this paper.

More detail comments are given below:

- The introduction section lists a large number of existing papers without discussing the relationship between these papers and this paper.

- The related work section has the same problem. The content of Page 1 to 8 are mostly not relevant to this paper.

- Section 3 (methodology) only provide a very superficial description of the manufacturing system components. It does not reveal any details on the technical challenges or design considerations of the physical and digital twins, e.g. the required accuracy of the digital twin for simulating the physical twin in the targeted scenarios, and how the system can guarantee the accuracy. Furthermore, the description of the systems is not replicable with the currently given details of the paper. The implementation of the DT in the paper is vague and does not exhibit a clear picture to the readers.

- Section 4 (results) basically shows the physical and digital twins works. As the research question (challenges) is not clear. It is hard to corelate the results with the contribution. For example, the result show manual loading time is longer than the time while using robot to load. Why is this result important in terms of the contribution? In addition, is this a commonly expected result while using robots?

- More description to the software used in the paper is required. Why are these software systems chosen?

- The full name of a acronym should be presented while they first appear, e.g. ERP, CNC.  

Author Response

- The introduction section lists a large number of existing papers without discussing the relationship between these papers and this paper.

- The related work section has the same problem. The content of Page 1 to 8 are mostly not relevant to this paper.

- Section 3 (methodology) only provide a very superficial description of the manufacturing system components. It does not reveal any details on the technical challenges or design considerations of the physical and digital twins, e.g. the required accuracy of the digital twin for simulating the physical twin in the targeted scenarios, and how the system can guarantee the accuracy. Furthermore, the description of the systems is not replicable with the currently given details of the paper. The implementation of the DT in the paper is vague and does not exhibit a clear picture to the readers.

- Section 4 (results) basically shows the physical and digital twins works. As the research question (challenges) is not clear. It is hard to corelate the results with the contribution. For example, the result show manual loading time is longer than the time while using robot to load. Why is this result important in terms of the contribution? In addition, is this a commonly expected result while using robots?

- More description to the software used in the paper is required. Why are these software systems chosen?

- The full name of a acronym should be presented while they first appear, e.g. ERP, CNC. 

Round 2

Reviewer 1 Report

The authors have performed a thorough revision of the manuscript after receiving the comments of the reviewers.

I consider now the paper suitable for publication.